# Evolutionary modification of AGS protein contributes to formation of micromeres in sea urchins

Jessica Poon[1], Annaliese Fries [1], Gary M. Wessel[1] & Mamiko Yajima [1]

Evolution is proposed to result, in part, from acquisition of new developmental programs. One such example is the appearance of the micromeres in a sea urchin that form by an asymmetric cell division at the 4th embryonic cleavage and function as a major signaling center in the embryo. Micromeres are not present in other echinoderms and thus are considered as a derived feature, yet its acquisition mechanism is unknown. Here, we report that the polarity factor AGS and its associated proteins are responsible for micromere formation. Evolutionary modifications of AGS protein seem to have provided the cortical recruitment and binding of AGS to the vegetal cortex, contributing to formation of micromeres in the sea urchins. Indeed, introduction of sea urchin AGS into the sea star embryo induces asymmetric cell divisions, suggesting that the molecular evolution of AGS protein is key in the transition of echinoderms to micromere formation and the current developmental style of sea urchins not seen in other echinoderms.

[1] MCB Department, Brown University, 185 Meeting Street, BOXG-L277, Providence, RI 02912, USA. Correspondence and requests for materials should be addressed to M.Y. (email: Mamiko_Yajima@brown.edu)

Asymmetric cell division is an important biological event that segregates developmentally important factors and diversifies cell fates. For instance, the *C. elegans* zygote iteratively divides asymmetrically starting with a large anterior and a smaller posterior blastomere, both with distinct cell fate determinants[1]. *Drosophila* embryonic neuroblasts divide asymmetrically to both self-renew and to generate the neurons of the larval nervous system[2]. In the mouse and chick, the neuroepithelium proliferates during neurogenesis by symmetric cell divisions, but then transitions to asymmetric cell divisions to produce self-renewing stem cells and neural precursor cells[3,4]. Introduction of asymmetric cell divisions into the developmental program is thus seen throughout phylogeny, and often plays essential roles to dramatically change the developmental program, which overall leads to morphological and functional diversification.

It is unclear, however, how these asymmetric cell divisions originally arose in the developmental program during evolution, and contributed to diversification. In this study, we address this fundamental problem using sea urchin micromeres as a model system. The first asymmetric cell division in the sea urchin embryo occurs at the 16-cell stage, yielding four macromeres and four micromeres. The micromeres are capable of inducing the site of invagination[5–7] and undergo yet another successive asymmetric cell division at the 32-cell stage to form the large and small micromeres. These two lineages autonomously result in two extremes of cell fate: the large micromeres remain inductive and develop the singular fate of skeletogenic cells for the larval skeleton[8], whereas the small micromeres result in the primordial germ cells[9,10]. The micromeres selectively accumulate a variety of transcription-, translation-, and signaling factors (e.g. *delta, wnt 8, pmar, vasa*) and the regulation of gene regulatory network of micromeres has been extensively studied in the field over the past two decades[11–18].

Importantly, this early asymmetric division that diversifies cell fates is seen only in echinoids (sea urchin, pencil urchins, and sand dollars) within echinoderms. The formation of the signaling center (micromeres) sufficient for inducing gastrulation is thus considered as a derived feature of the phylum. It is, however, still largely unknown what mechanisms initiated formation of this lineage through the asymmetric cleavages. Here, we report that Activation of G-protein Signaling (AGS), a polarity factor, regulates the asymmetric cell division resulting in formation of the micromeres with organizing center activity in the sea urchin embryo. We demonstrate that a molecular modification of the AGS protein contributes to the evolutionary transition in this important developmental mechanism during echinoid diversification, providing a paradigm for how cell types arise in development during organismal evolution.

## Results

### Key signaling molecules enrich prior to micromere formation.
The process of asymmetric cell division during the 8–16 cell stage transition in the sea urchin can be parsed into three steps, (1) movement of the nuclei in the vegetal-tier blastomeres towards the vegetal pole at the 8-cell stage[19], (2) asymmetrically shifted spindle positioning and unequal-sized cytokinesis by anchoring one of the centrosomes to the vegetal cortex[20], and (3) enrichment of factors critical for the organizer function into the micromeres[21–23]. In this study, we focus on the third step, and use Vasa (Supplementary Table 1) as a metric of molecular enrichment in micromeres through the asymmetric cell division. Vasa mRNA and protein are present maternally and remain in every blastomere until the 8-cell stage, when it becomes specifically enriched in the micromeres, and subsequently into the small

micromere lineage with two successive asymmetric cell divisions. We found that the Vasa protein and other factors that function in micromeres[11,12,17,24] asymmetrically accumulate in the vegetal-most side of the spindle and its surrounding cytoplasm during M-phase, prior to the asymmetric cytokinesis (see Supplementary Table 1 for listing of factors, Supplementary Fig. 1a, b). To determine if the asymmetric cell division is linked to the differential distribution of the micromere factors, we first disrupted micromere formation by SDS treatment[5,6,19,23]. This treatment randomized axes of cell division and results in equal-sized daughter cells at the 8–16 cell stages (Supplementary Fig. 1c). Importantly, Vasa was no longer asymmetric in the treated embryos lacking an asymmetric cell division at the vegetal pole. We also found that the conserved polarity factors AGS (LGN/Pins as orthologs)[25] and Gαi[26] lost its normal asymmetric accumulation to the vegetal pole (Supplementary Tables 1–3; Supplementary Fig 1d; see following sections for details). These results show that asymmetric cytokinesis and enrichment of specific molecular factors at the vegetal pole are linked.

### AGS/Gαi are essential for asymmetric cell divisions.
The sea urchin has only one AGS gene and it is highly similar to both the human AGS3 and to the human LGN (Supplementary Table 2). Since human AGS3, but not LGN, replicated the localization of endogenous sea urchin AGS (Supplementary Fig. 2a), we refer to the sea urchin sequence with functional similarity as AGS, and not LGN. We found that AGS was transiently localized at the vegetal cortex beginning at the 4–8-cell stage and maintained in this highly localized position until the 32-cell stage but only in the asters after late embryonic stage (Supplementary Fig. 2b, c). Gαi, a partner protein of AGS, showed an even more restricted domain in the vegetal cortex (Supplementary Fig. 2d).

To test if AGS/Gαi is functionally involved in the unequal cell division, we knocked down (kd) these molecules either by morpholino antisense oligonucleotides (MO) and/or by specific chemical inhibitors. Morpholinos were used here as the best post-transcriptional inhibition of specific protein accumulation as CRISPR/Cas9 targeted gene inactivation would not be effective at this early embryonic stage[27]. The AGS-MO and Gαi-MO successfully diminished the cortical AGS and Gαi signal, respectively, in the cortex, and these diminished AGS and Gαi signals appeared to be associated with loss of asymmetric cell divisions (Fig. 1a, b and Supplementary Fig. 3a).

In both AGS-kd and Gαi-kd embryos, the first step in asymmetric cell division was blocked, that is, movement of the nucleus to the vegetal cortex (Fig. 1c, d). The orientation of the spindle was also altered in AGS-kd cells, but less so in Gαi-kd, whereas the asymmetric spindle localization to the vegetal cortex was perturbed in either knockdown through the loss of centrosome anchoring (or vegetal astral growth at the vegetal cortex) (Fig. 1e, f). This specific molecular perturbation resulted in more even-sized daughter cells (Fig. 1g, h). In AGS-kd cells, the selective enrichment of Vasa in the vegetal cortex was lost during M-phase (Fig. 1i, j) as well as at the 16-cell stage (Fig. 1k), as was the enrichment of other developmentally important molecules (Supplementary Fig. 3b). In Gαi-kd or –inhibitor (PTX)-treated cells, the effect on unequal cell division was rather mild yet Vasa enrichment in the vegetal cortex was still significantly reduced (Fig. 1i, white dashed circles). Importantly, MOs at the same concentration that were irrelevant to the asymmetric mechanism had no effect on micromere formation nor Vasa distribution at the 16-cell stage (controls in Fig. 1 and Supplementary Fig. 3c; see also the Methods). Taken together, these results suggest AGS may be the key factor regulating the micromere formation.

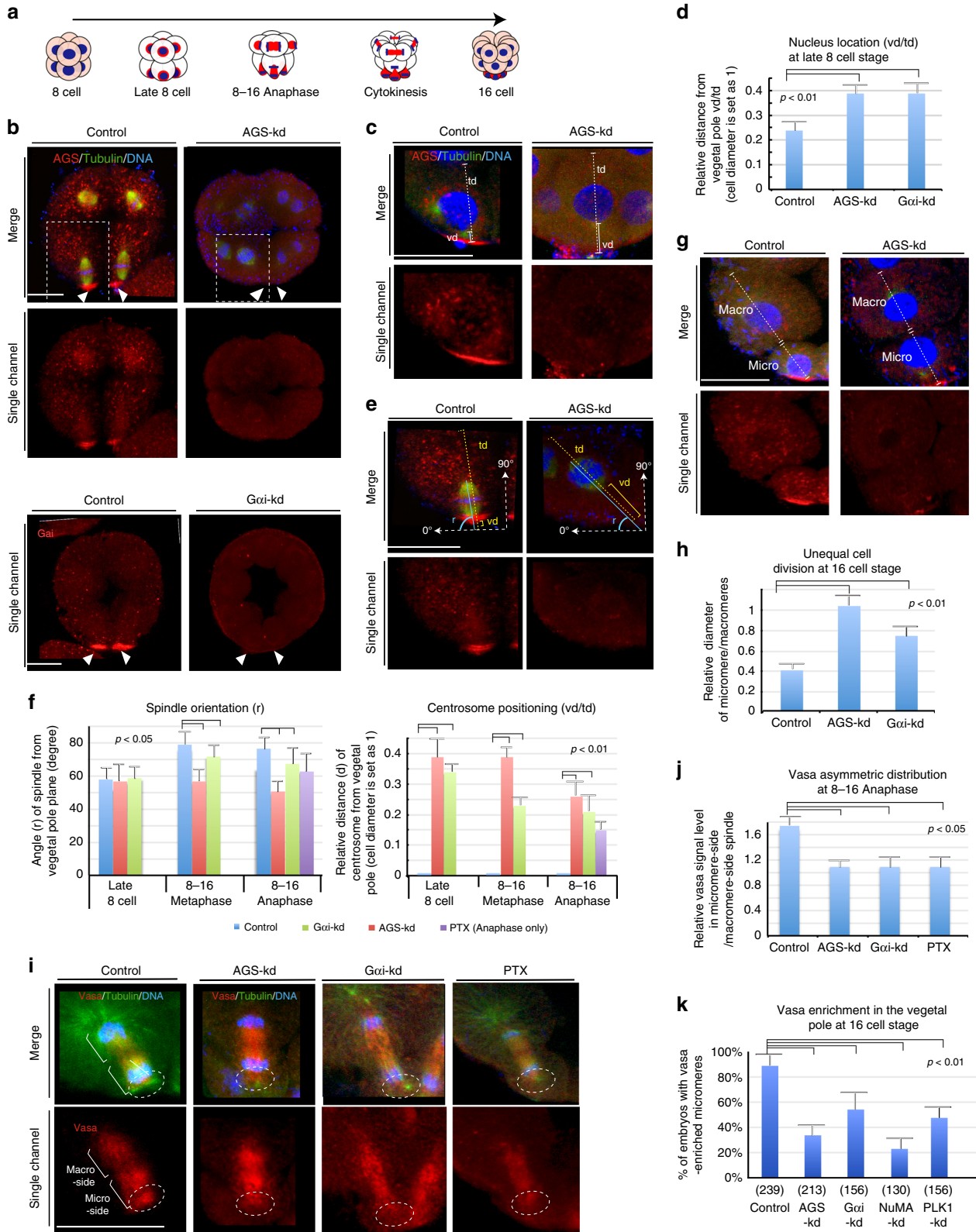

## AGS-mediated mechanism regulates micromere formation.

AGS orthologs are thought to control spindle positioning by forming a complex with microtubules and associated proteins such as NuMA and Dynein (Supplementary Table 1)[28,29]. Further, it is reported that this interaction is negatively regulated by Polo-like kinase 1 (PLK1) in HeLa cells (Supplementary Table 1)[30], providing stringent controls both for symmetric and asymmetric

cell division. To determine how the AGS-containing complex contributes to micromere formation and asymmetric Vasa distribution, we examined interactions between these core proteins involved in asymmetry. We first performed Gαi immunoprecipitation (IP) using 8–16 cell stage lysates in the presence or absence of a PLK1 inhibitor (BI2536) or a recombinant PLK1 protein (Fig. 2a, b). We found that the Gαi-IP pulled out Vasa

**Fig. 1** AGS and Gαi are necessary for asymmetry in cell division and Vasa localization. **a** Diagrammatic representation of the transition from the 8–16 cell stage. Nucleus, Blue; Vasa, Red. The major organizing center of this embryo is the bottom-most (vegetal) cells, the micromeres. **b** AGS-kd (upper panels) result: In the control (2 mM Nanos-MO stock), the vegetal side of the aster is near to the vegetal cortex (arrowheads), whereas AGS-kd (2 mM AGS-MO stock) shifted the mitotic apparatus to the center of a cell (arrowheads). White squared regions are enlarged in **e**. Gαi-kd (lower panels) result: Gαi-kd diminished Gαi signal substantially at the vegetal cortex (arrows). **c–j**, Representative images and graphs for each measurement are shown. Each measurement was performed on 10 individual embryos that showed no AGS or Gαi-signal at the cortex as described in 'Data analysis' of the Methods. **c**, **d** Migration of nucleus to the vegetal cortex was repressed by AGS or Gαi-kd (measured by the relative value of vd/td). **e**, **f** Spindle orientation (r) and centrosome positioning (vd/td) was altered by AGS-kd. **g**, **h** Unequal cell division (micromere formation) was inhibited by AGS or Gαi-kd (measured by the relative value of Micro/Macro). **i**, **j** Asymmetric distributions of Vasa to the micromere-side of the spindle was diminished by AGS or Gαi-kd (measured by the relative value of Micro-side/ Macro-side). Both AGS- and Gαi-kd as well as PTX that locks Gαi in inactive state diminished the flatten-shaped micromere-side asters as well as Vasa signal from the vegetal cortex (dashed-circles), resulting in symmetric localization of Vasa over the spindle. Scale bars = 20 µm. **k** Proportion of embryos that formed micromeres and showed enriched Vasa signal at the 16-cell stage. () indicates total number of embryos counted. The statistical significance obtained by one-way Nova between controls and experimental samples was indicated as p-value. Each experiment was performed at least three independent times. For a detailed procedure of each data analysis, please refer to the "Data analysis" section of the Methods. Scale bars = 20 µm

both in normal embryos and in the absence of PLK1 activity, but Vasa did not interact with Gαi in the presence of PLK1 activity. Gαi also lost its interaction with sea urchin NuMA-like protein (suNuMA; Supplementary Fig. 4b; see also the Methods for details) and Dynactin (p150) in the presence of PLK1 activity (Fig. 2a, +PLK1). IgG-IP as a control was performed alongside of the Gαi-IP experiments, and did not pull down any specific proteins detected by these antibodies with or without PLK1 activity (Supplementary Fig. 4a). Similar results of Gαi interactions were seen in the presence of Nocodazole that destabilizes microtubules, in which AGS remained associated with Gαi (Fig. 2b). Vasa also associated with Gαi and suNuMA in vivo independent of microtubules at the vegetal cortex (Fig. 2c–e), consistent with the protein interaction studies above. Further, the interaction among suNuMA–Dynein–Vasa proteins was tested by Vasa-IP in the presence or absence of PLK1 (Fig. 2f). In the presence of PLK1, Vasa interaction with Dynein and suNuMA was significantly reduced. These results suggest that the cortical Gαi-AGS is anchored to a suNuMA–Dynein–Vasa complex at the vegetal cortex, but this complex is dissociated in the presence of PLK1 kinase activity. Overall, these results suggest that PLK1 leads to dissociation of the Gαi–AGS–NuMA–Dynein (GAND)–Vasa complex and could contribute to an asymmetric distribution of Vasa on the spindle.

To test this finding in vivo, PLK1 distribution was analyzed in the intact embryo. With multiple different PLK1 antibodies (Supplementary Table 4), we detected PLK1 evenly distributed over the spindles of every blastomere, which were reduced in the vegetal side of the spindle during micromere formation (Fig. 3a, arrows; Supplementary Fig. 4c). This signal was seen with antibodies that recognize the middle region of PLK1, as well as antibodies that bind to the C-terminus of PLK1/PLK1L. This same profile was less distinct though when live imaging with PLK1/PLK1L-GFP perhaps as a concentration dependent mechanism (Supplementary Fig. 5a). To test directly if PLK1 activity at the vegetal cortex inhibits GAND-complex activities and micromere formation, membrane-targeted PLK1 was introduced in the embryos. As expected, over 85% of these embryos formed abnormally large-sized micromeres with no apparent Vasa enrichment nor cortical Dynactin localization ($n = 166$) (Fig. 3b and Supplementary Fig. 5b, c, membrane-mCherry-PLK1). In contrast, embryos expressing a membrane-targeted PLK1-mutant that lacks kinase activity (membrane-mCherry-PLK1-dead) showed no abnormal phenotype, and only 13% of the embryos failed in micromere formation ($n = 115$) (Fig. 3b and Supplementary Fig. 5b, c, membrane-mCherry-PLK1-dead). These results suggest that reduced PLK1 kinase activity at the vegetal cortex might be important for the GAND-complex stabilization and micromere formation. To further test this

premise, we treated embryos with PTX (Gαi inhibitor) or Ciliobrevin A (Dynein inhibitor)[31] for 10 min during M-phase of 8–16 cell stage (Supplementary Fig. 5d, arrows), and found that PLK1 was expanded to the vegetal-spindle pole region. The GAND complex may, therefore, regulate PLK1 activity to recruit the spindle pole to the vegetal cortex (Fig. 3c, d). This model is consistent with the AGS/Gαi-kd phenotype that failed in asymmetric cell division as well in Vasa enrichment at the vegetal cortex. Based on these observations, we propose that Vasa (and potentially other regulatory molecules) directly binds to the GAND complex that is stabilized at the vegetal cortex during asymmetric cell division, and which results in the formation of a distinct cell lineage, the micromere (Fig. 3e).

**AGS-directed mechanisms regulate specification of micromeres.** Based on the above model (Fig. 3e), the AGS and GAND complexes may regulate not just asymmetric cell division but also the cell fate of micromeres. Micromeres are known to function as organizers in this embryo and signal to adjacent cells for axial induction required for gastrulation[6,7]. A reduced knockdown of AGS (1 mM stock or lower) often resulted in formation of micromere-like cells (although the size was typically larger), and the resultant embryos developed into blastulae. Most of them, however, failed to develop endoderm, a major cell fate resulting from organizer function of the micromere, and these embryos showed no expression of the endoderm marker Endo 1(Fig. 4a)[32]. This defect in micromere formation and endoderm induction was rescued by introduction of a MO-insensitive AGS mRNA (Fig. 4b, c). These results indicate that embryos disrupted with AGS/Gαi expression lacks not just morphological and but also functional micromere-like cells at the 16-cell stage. Therefore, proper molecular segregation during micromere formation might be the key for organizer function. To be noted, continuous AGS-kd in these embryos showed more severe phenotypes than pulse-inhibition (10 min at the 8–16 cell stage) by SDS[23], likely by preventing the embryos from re-regulating the developmental program.

To test more directly if developmental failure of the AGS-kd embryos was due to loss of this signaling center, or to disruption elsewhere in the embryo, we transplanted normal micromeres (with native inductive capability and labeled with RITC) to host AGS-kd embryos at the 16-cell stage (Fig. 4d). When the resultant embryos were examined at Day 3 post-fertilization (PF), 75% (9/12) of embryos that were RITC-positive and thereby chimeric, successfully gastrulated and developed into normal-looking larvae by Day 3 PF (Fig. 4e). We conclude that the major developmental deficiencies of the AGS-kd embryos are a result of a hypo-inductive field at the vegetal pole. This suggests that AGS-directed asymmetric cell division may be critical for both asymmetric

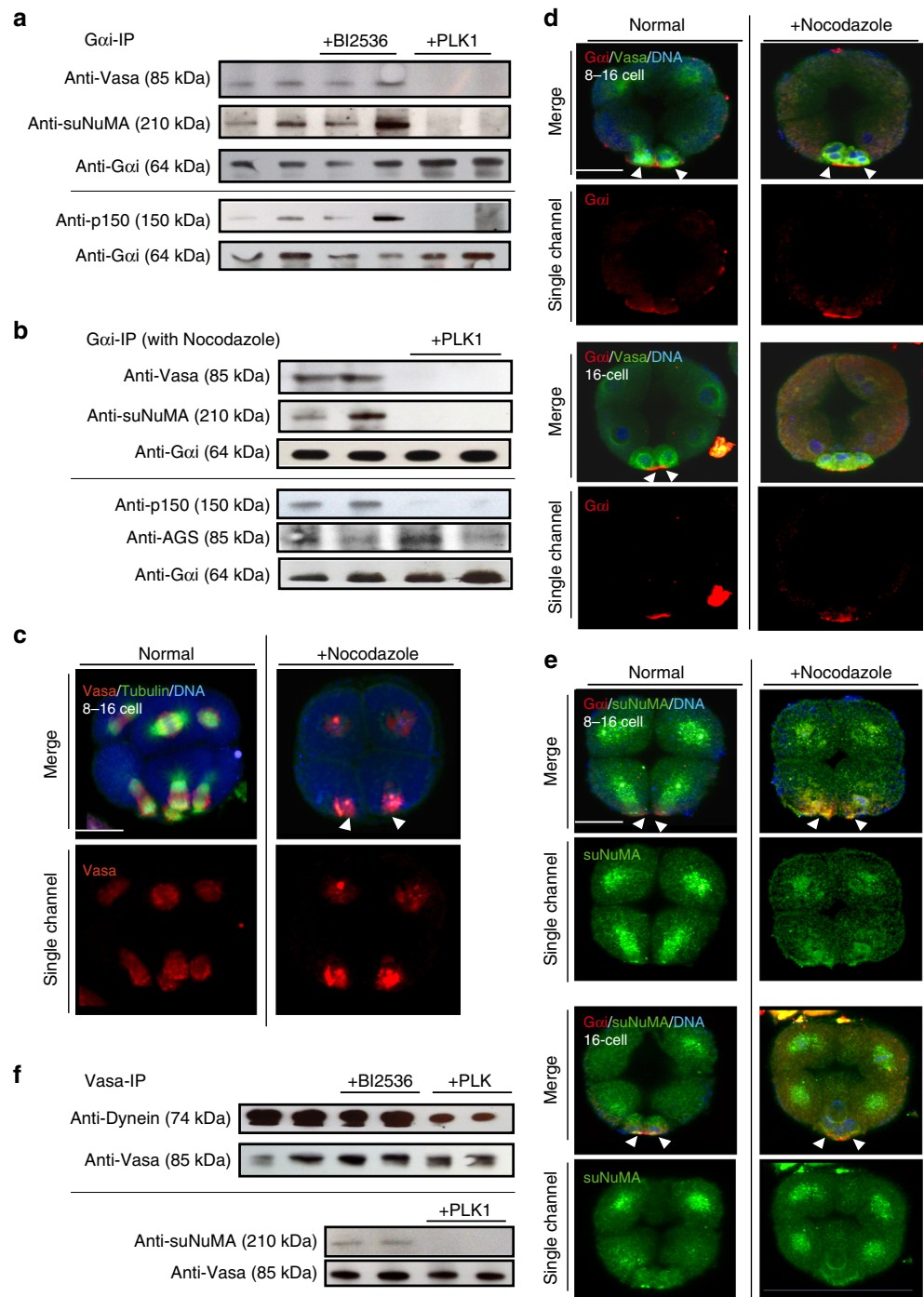

**Fig. 2** Stability of the regulatory protein complex at the vegetal cortex is sensitive to PLK1 activity. **a** Gαi-IP results. suNuMA, AGS and dynein interact with Gαi in extracts from the 8–16 cell stage. **b** Gαi-IP was performed after treatment with Nocodazole to destabilize microtubules. **c–e** Embryos were treated with Nocodazole at 8–16 cell stage for 10 min to disturb microtubule polymerization. **c** Nocodazole diminished spindle but Vasa (red) remained asymmetrically localized at the vegetal cortex independent of microtubules. **d** Vasa (green) remained at the close proximity to Gαi (red) after the treatment (arrowheads). **e** suNuMA (green) also remained linked to the Gαi (red) signal at the vegetal cortex after the treatment with Nocodazole (arrowheads). Scale bars = 520 μm. **f** Vasa-IP results. Vasa interacts with dynein and suNuMA in the absence of PLK1. In a, b, f, IPs were conducted at least three independent times and two each of the IP-ed samples are shown. Each of the top and bottom groups of immunoblots (demarcated by the line) was processed from a single blot. () indicates the molecular weight of each protein. The representative phenotypes of 70% or larger in each population are shown (*n* = 30 or larger)

cytokinesis and for specification of the micromere's inductive capabilities. To be noted, the reverse experiment of transplanting AGS-kd micromeres to normal embryos was not possible in this study since micromeres did not form morphologically in AGS-kd embryos.

**AGS draws spindle poles to the cortex.** Based on the above observations, AGS appears to be a key factor both for micromere formation and function. We hypothesize that AGS (and/or Gαi) directly recruits the spindle pole to the cortex for anchoring, and allows enrichment of the fate determinants through interaction of

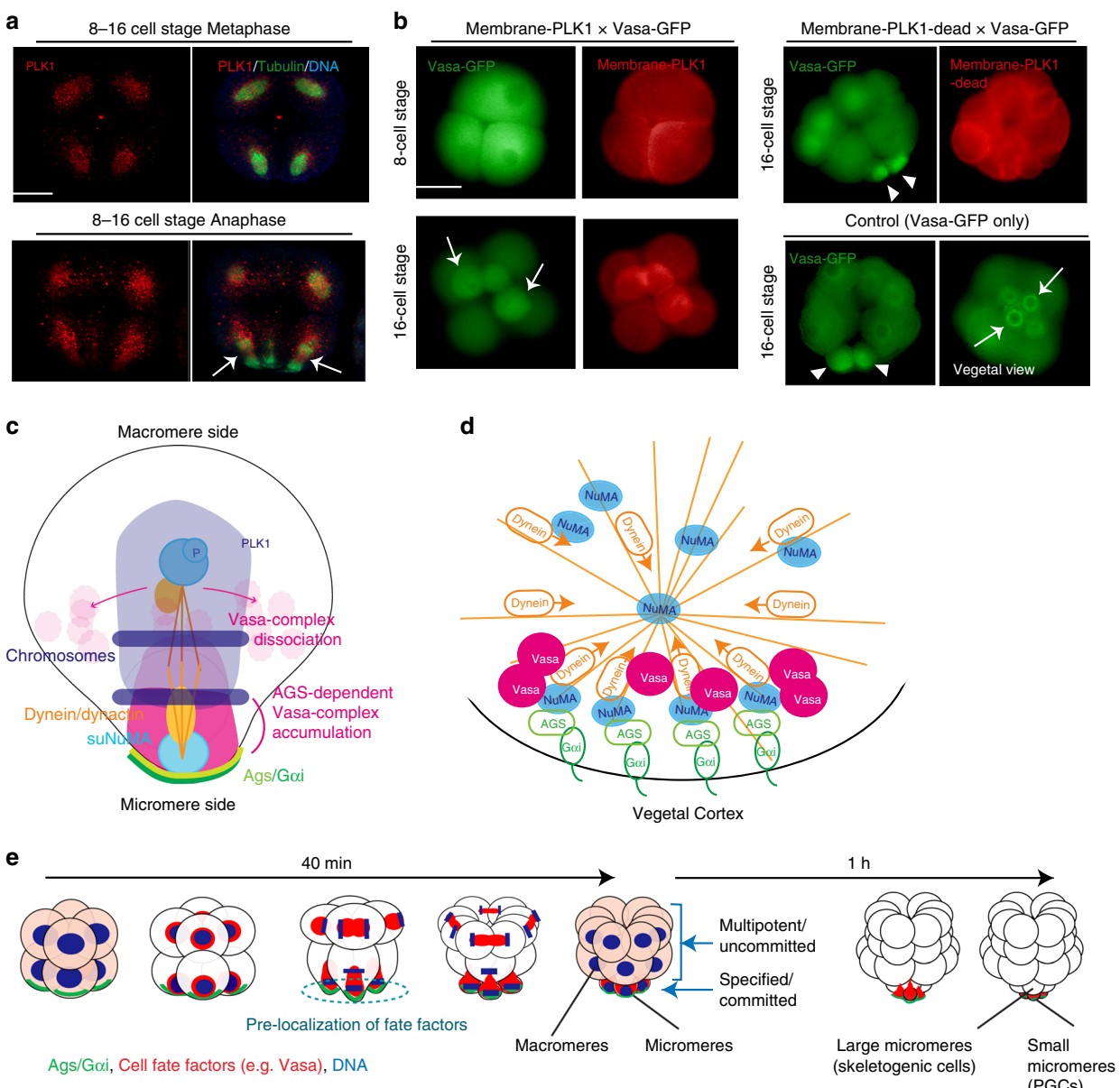

**Fig. 3** PLK1 activity alters Vasa and complex localization at the vegetal cortex and a model for AGS-dependent micromere formation. **a** PLK1 was localized over every spindle yet was depleted from the vegetal side of the 16-cell spindle (arrows). **b** membrane-mCherry-PLK1 mRNA (1 μg/μl stock) was co-injected with Vasa-GFP mRNA (1 μg/μl stock) and images were taken in live embryos. These embryos either formed no micromeres, or resulted in the formation of aberrant size and number (1–3 cells) of micromere-like cells at the 16-cell stage (arrows), whereas embryos with membrane-mCherry-PLK1-Kinase-Dead (PLK1-dead) or Vasa-GFP mRNA only showed no knockdown phenotype and formed micromeres with Vasa enrichment (arrowheads). For **a–b**, the representative phenotypes of 85% or larger in each population are shown (n = 100 or larger). **c** A working model for the molecular mechanisms of micromere formation and Vasa distribution in normal or Gαi/AGS perturbed embryos. Diagram shows a vegetal tier of blastomeres at 8–16 cell stage that is about to form the micromeres in this embryo. The amount of AGS on the centrosomes is significantly lower at this stage and omitted from this diagram. **d** A hypothetical model of molecular organization at the vegetal cortex. **e** A summary diagram of AGS-dependent asymmetric cell divisions. AGS-dependent asymmetric localization of cell fate factors enables a rapid lineage segregation and transition to the micromere specification pathway by pre-localizing organizer factors to the vegetal cortex during asymmetric cell divisions

the GAND complex. To test this hypothesis, we induced cortical localization of AGS in the embryo by overexpressing AGS, which successfully induced the cortical AGS localization in every blastomere (Fig. 5a, arrowheads). In these embryos, many of the microtubules were recruited to the cortex at M-phase, which made spindle morphology, cell division patterns, and morphology of the embryo abnormal (Fig. 5b, c), preventing proper micromere formation (Supplementary Fig. 6a; Supplementary Movie 1). We noted that blastomeres enriched in AGS and Gαi dissociated from the embryo by yet unknown reasons (Supplementary

Fig. 6b, c, arrows). Further, AGS overexpressing embryos often showed more than two spindle poles (Supplementary Fig. 6d, arrowheads). With Gαi overexpression, on the other hand, spindle morphology was normal and the spindles were never recruited to the cortex (Supplementary Fig. 6d, Gαi-OE). Spindle orientation was, however, altered, resulting in symmetric cell divisions at the 16-cell stage: spindles of the vegetal blastomeres appeared to be oriented horizontally, relative to the A-V axis, rather than vertically (Supplementary Fig. 6c, Gαi-OE). Therefore, Gαi might play an important role in spindle orientation.

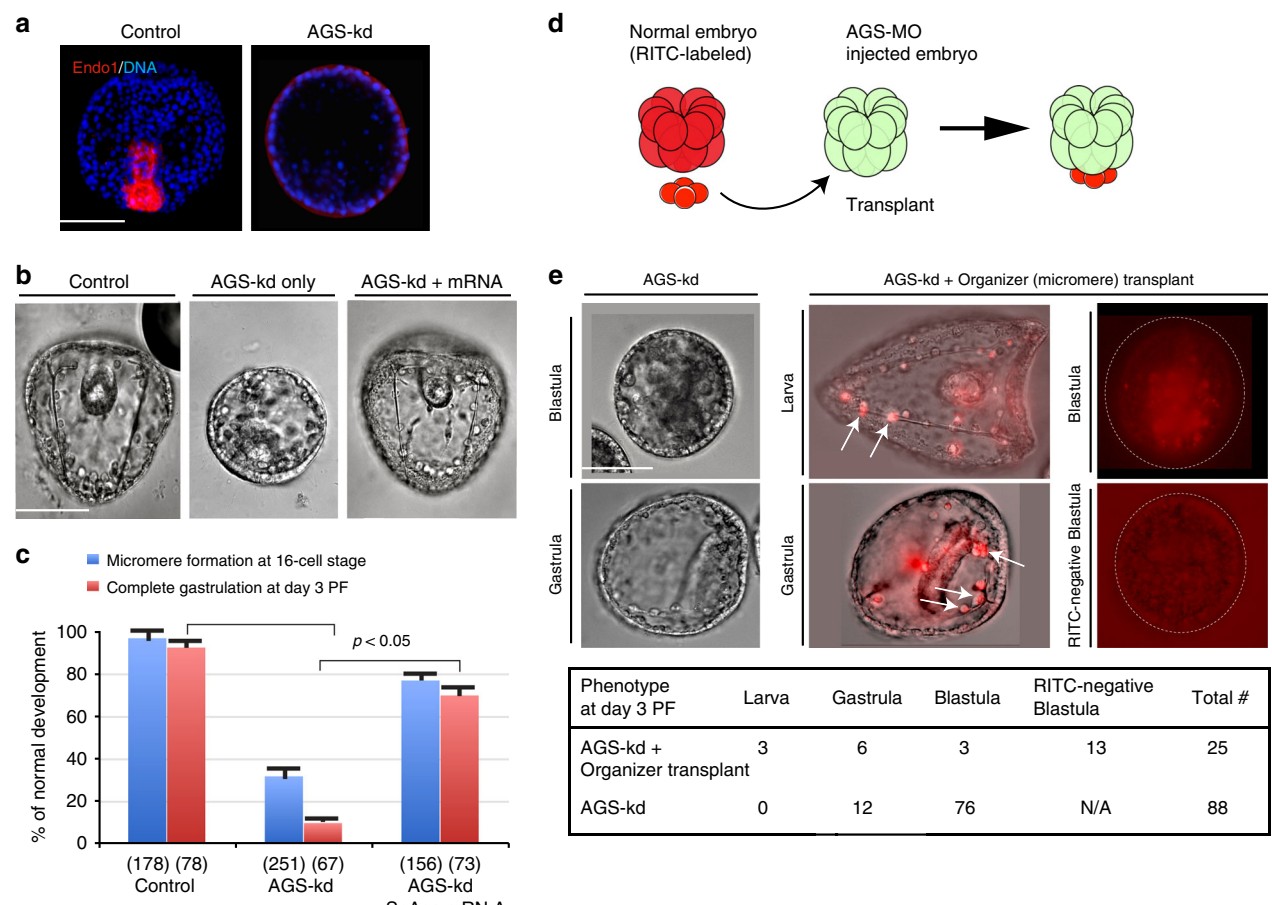

**Fig. 4** AGS is important for micromere's function as organizers. **a** An endoderm marker, Endo1 (red) was highly expressed in the archenteron of gastrula (Day 2 PF) in the normal embryo, whereas no archenteron or Endo1 expression was found in AGS-MO injected embryos even with lower dose (1 mM). DNA, blue. The representative phenotypes of 85% or larger in each population are shown ($n = 50$ or larger). **b**, **c** AGS-MO (1 mM stock) injection resulted in embryonic death after blastula stage (AGS-kd only) yet co-injection of SpAgs-mRNA (0.25 μg/μl) that is insensitive to AGS-MO rescued the knockdown phenotype (AGS-kd + mRNA) to the level of Control embryos (1 mM Nanos-MO stock). The graph in c demonstrates % of embryos that formed micromeres at 16-cell stage (blue bars) and completed gastrulation at Day 3 PF (red bars). () indicates the number of embryos examined. Each experiment was performed at least three independent times. The statistical significance obtained by one-way Nova is indicated as *p*-value. Scale bars = 50 μm. **d**, **e** AGS is essential for specification of micromeres as a developmental organizer. Diagram indicates the experimental procedure of micromere-transplant. Over 86% of embryos injected with AGS-MO (2 mM stock) remained as blastula and failed in gastrulation (AGS-kd). On the other hand, when these host embryos received wild-type micromeres (AGS-kd + Organizer (micromere) transplant), the RITC-labeled micromeres rescued the developmental defects, and differentiated normally into skeletogenic and coelomic pouch cells (red, arrows). In some cases, however, the transplanted cells entered the blastocoel and the embryos failed in gastrulation (a dashed circle in Blastula). Table in e indicates the number of embryos that successfully developed to each developmental stage (Larva, Gastrula, or Blastula), or that showed no detection of RITC-labeled transplanted cells (a dashed circle in RITC-negative Blastula), or that were observed in total (Total #). Scale bars = 50 μm

These results suggest that AGS might be a dominant factor that directly recruits microtubules and spindle poles to the cortex, and by doing so at the vegetal cortex, directs micromere formation.

**Evolution of AGS protein function altered the cell division plane**. The sea urchin (echinoid) is the only class of echinoderms that undergoes early asymmetric cell divisions to form micromeres with the functionality of an organizing center. Coincident with micromere formation in phylogeny is an early lineage restriction of a larval skeleton and primordial germ cells by the 32-cell stage[8,10]. In contrast, other echinoderm taxa all undergo symmetric cell divisions, and do not have distinct organizer functions within early blastomeres. Micromeres are thus considered as a derived character in echinoderms[33,34].

Considering the essentiality of AGS in asymmetric cell division, we hypothesized that evolutionary modifications of AGS in

echinoids might have played a role evolutionarily in introducing micromeres at the 16-cell stage. To test this hypothesis, we analyzed AGS and Gαi expression in two other echinoderms. One is the sea star (*P. miniata*) that is considered to retain the ancestral developmental characteristics of echinoderms, and the other is the pencil urchin (*E. tribuloides*) that contains characters intermediate to sea stars and sea urchins during embryonic development[34–37]. Importantly, no signal of AGS nor Gαi localization was found in the vegetal cortex in sea star embryos even though we thoroughly examined over 50 embryos (Fig. 6a), suggesting a distinct mechanism for spindle orientation than seen in the asymmetric cell division of sea urchins.

Sea star AGS (PmAGS) and sea urchin AGS (SpAGS) protein sequences are nearly identical except in the C-terminus in which the GoLoco motifs are located (Supplementary Table 5). This alignment is consistent in the nine echinoderm species tested throughout the phylum (Supplementary Table 6). The

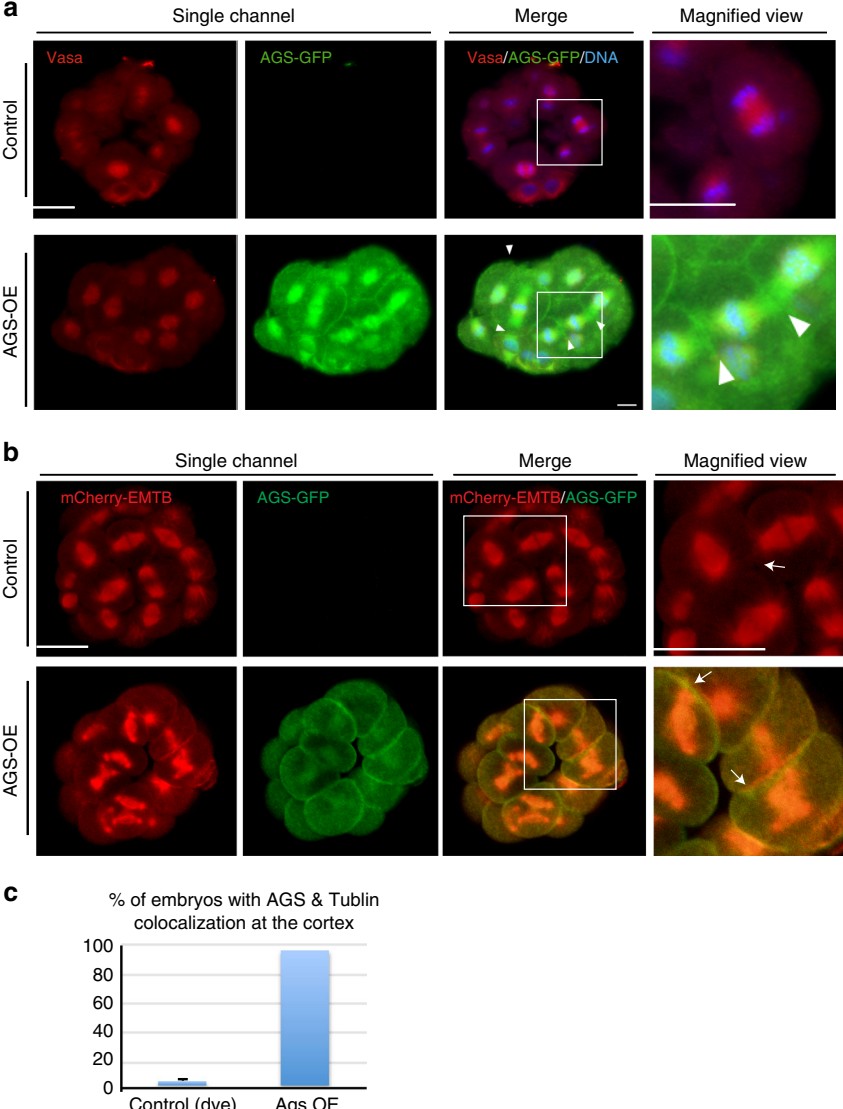

**Fig. 5** Overexpression of AGS changes spindle orientation and anchoring. **a** AGS-GFP overexpression (AGS-OE; 1 μg/μl) changed overall embryo morphology at the 16-cell stage. Images were obtained by immunofluorescence with methanol fixation that diminishes the live GFP signal. The AGS-GFP signal was thus visualized by anti-GFP antibody. At M-phase, spindle orientation was randomized and many of the spindle poles were anchored to the nearby cortex (arrowheads in the magnified view). ~50% of embryos showed phenotype, n = 14. **b, c** AGS-OE caused ectopic recruitment of microtubules labeled by mCherry-EMTB to the cortex (arrows) in over 80% of embryos, n = 25. Images taken are from live embryos. Each experiment was performed at least three independent times. Scale bars = 20 μm

N-terminus of AGS or LGN has been reported to be important for the molecule's diverse functions such as regulation of spindle orientation through its interaction with NuMA and the C-terminal GoLoco motifs for the cortical association of AGS/LGN through the interaction with Gαi[38,39]. In pencil urchins, embryos typically form 0–4 micromere-like cells. AGS/Gαi localization and Vasa enrichment were detected at the cortex and in the cytoplasm of the vegetal most cells, but only when micromere-like cells were formed in the embryo (Fig. 6b–d, arrows), supporting that cortical AGS/Gαi localization and Vasa enrichment are all closely linked with each other.

**SpAGS induces asymmetric cell division and vegetal polarity.**
Through the motif prediction of each echinoderm's AGS molecule, it appears that GoLoco motif #1 is present only in echinoid species (pencil urchin, sea urchin and sand dollar), which results in extra GoLoco motifs present in the C-terminus of echinoids

(Supplementary Table 7a). Since the GoLoco motif #1 has been reported to be crucial for Pins' recruiting activity through Gαi to the cortex in *Drosophila*[39], we speculated that appearance of this GoLoco motif #1 in the sea urchin AGS may have provided an increased association with the cortex. To test this, we introduced sea urchin AGS (SpAGS) mRNA into the sea star embryo that normally undergoes *symmetric* cell division at the 8–16 cell stage (Fig. 7a). Remarkably, ~80% of these embryos underwent random asymmetric cell divisions from the 2–16 cell stages. Among those, approximately 15% of them formed micromere-like cells when expressing the sea urchin AGS. (Fig. 7b, c, arrows), resembling the 16-cell stage of the sea urchin embryo. On the other hand, in negative control embryos injected with SpAGS that lacks the GoLoco motif #1 (AGS-dGoLoco1) or the entire C-terminus including GoLoco motifs #1–4 (AGS-dC-term) or dye, no significant phenotypic alteration was observed (Fig. 7b, c and Supplementary Fig. 7a). In the AGS-overexpressing (AGS-OE) embryos, AGS was enriched in the entire cortex as well as

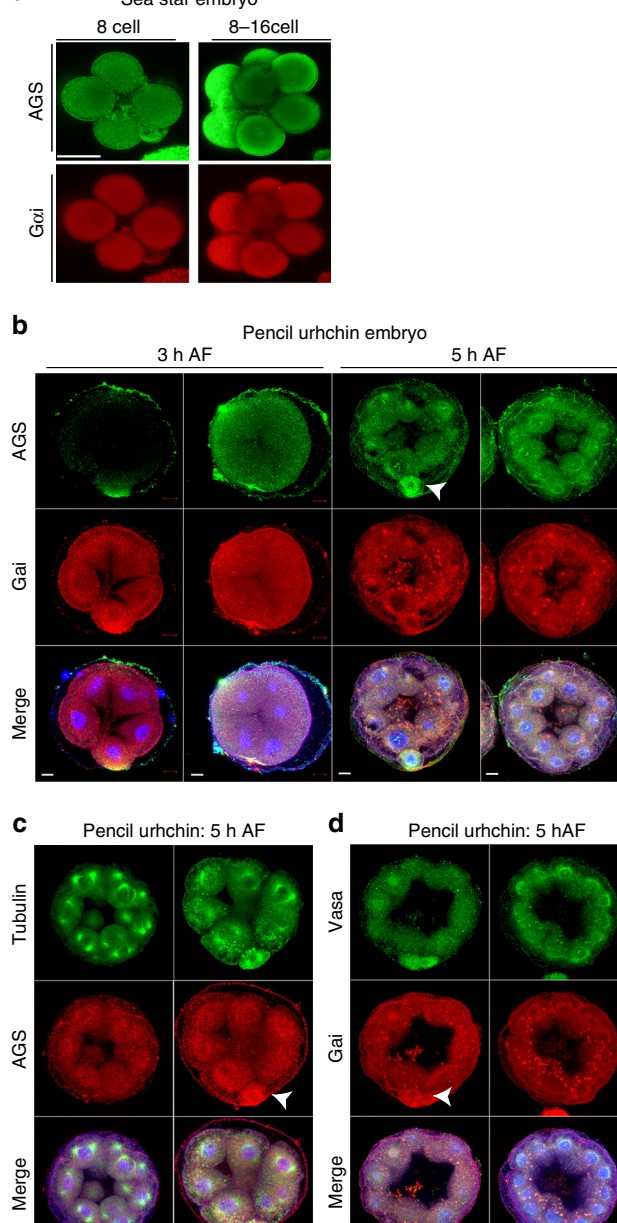

**Fig. 6** AGS, Gαi, and Vasa proteins are enriched in micromere-like cells of the pencil urchin embryo but not in the sea star embryo.
**a** Immunofluorescence images of AGS (green) and Gαi (red) in sea star embryos. No specific signal was found in any region of the cortex during early embryogenesis in over 90% of embryos, $n < 50$.
**b**, **d** Immunofluorescence images of pencil urchin embryos. In **b**, at 8 and 16 cell stages, embryos were enriched in AGS and Gαi at the vegetal cortex (arrows). These proteins were also enriched within the cytoplasm of the vegetal blastomeres or micromere-like cells in some embryos. Embryos lacking micromere-like cells also lacked AGS and Gαi enrichment on the cortex. $n = 20$. In **c** AGS was localized to the centrosomes in addition to the cortex, similar to that of sea urchin (*S. purpuratus*). $n = 10$. In **d**, Vasa (green) was enriched in the cytoplasm of the micromere-like cells, and enrichment was absent when micromere-like cells were not present. $n = 15$. The representative phenotypes of 80% or larger in each population are shown. Each experiment was performed at least two independent times. Scar bars = 10 μm

cytoplasm and/or on the spindle during M-phase as seen in the AGS-OE sea urchin embryo (Fig. 7d, arrow). After blastula stage, these embryos displayed extra sites of epithelial invaginations, a typical phenotype induced by the organizer's activity in sea urchins (Fig. 7e, f, arrowheads). These results suggest that sea urchin AGS indeed has an ability to induce asymmetric cell divisions and potentially a polarity-inducing activity even in the sea star embryos, a distantly related echinoderm.

In these AGS-OE embryos, however, Vasa was not significantly enriched in micromere-like cells (Supplementary Fig. 7b, arrow) nor in the germline at late larval stage (Fig. 7e, arrowheads). This may be due to universal overexpression of SpAGS, randomizing the polarity in the entire embryo. To clarify this possibility, we tracked the lineage of micromere-like cells by injecting red fluorescent dextran (dye) at 16–32-cell stage (Fig. 8a). In echinoderm embryos, it is well established that vegetal blastomeres contribute to endomesodermal lineages whereas animal blastomeres contribute to ecto-mesodermal lineages. As a result of this lineage tracing, the majority (~60%) of micromere-like cell descendants were found in the endo-mesoderm, especially in the future hind gut (an intestinal region of the gut) (Fig. 8b, arrows), whereas only ~20% of them were found in ectoderm (Fig. 8c). Similar results were also seen by using the photoconvertible protein Kaede as a lineage tracer (Supplementary Fig. 7d-e). These two independent experiments consistently suggest that micromere-like cells create a vegetal fate.

These findings led us to consider two possible explanations: either SpAGS has increased affinity to the vegetal cortex of the sea star and induces micromere-like cell formation preferentially at the vegetal pole, or SpAGS facilitates a new vegetal polarity as micromeres do so in the sea urchin embryo. To distinguish between these possibilities, we injected SpAGS mRNA randomly into one of 8-cells during the 8–16 cell stage transition (Fig. 8d). Control embryos injected with dye only resulted in the expected distribution of signal between cells of the endoderm, mesoderm, and ectoderm (Fig. 8e, f). In the experimental group, on the other hand, SpAGS-injected cells largely contributed instead to endoderm, and in particular, to the future hind gut of the endoderm (Fig. 8e, arrows; 8g). The embryos with SpAGS signal exclusively in the endoderm developed a single gut and appeared rather more normal than the universally SpAGS-overexpressing embryos (e.g. AGS-OE in Fig. 7d), while embryos with signal both in the ectoderm and endoderm showed multiple invaginations. These results support the premise that SpAGS facilitates a vegetal polarity even in the sea star embryo, in a similar manner that micromeres induce the vegetal polarity in the sea urchin embryo, and that this functional transition can occur even after the 8-cell stage. Based on these observations, SpAGS appears to be sufficient to induce asymmetric cell divisions, and potentially to facilitate the vegetal polarity even in the sea star embryo.

## Discussion

Acquisition of an asymmetric cell division during development may initiate a cascade of events leading to an inductive center in development (summarized in Fig. 3e). In the sea urchin embryo, it has been proposed that orientation of early embryonic cell divisions is primarily determined by the elongated shape of the cell but independent of cortical effectors[40,41]. Our findings in this work suggest that this is not the case when the embryo undergoes asymmetric cell division: the cortical factors play major roles in micromere formation. The first symmetry-breaking event in this embryo is the horizontal cell division that occurs at the 4–8 cell stage. Coinciding with this division, AGS/Gαi localization is initiated at the vegetal cortex. AGS/Gαi knockdown by MO

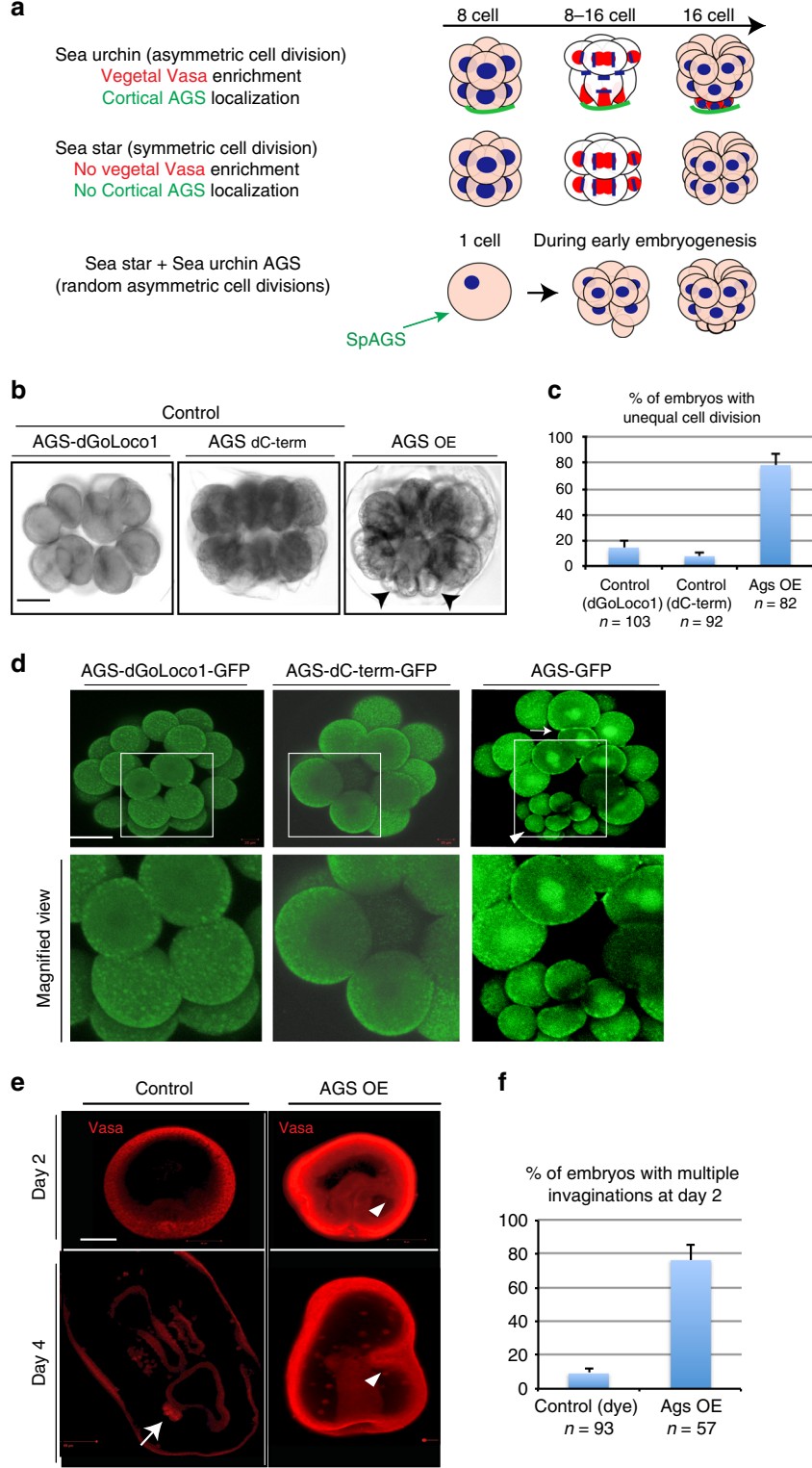

injection, however, showed no effect on this first horizontal cleavage in this study, which may be due to contributions of maternal AGS/Gαi proteins present in this early stage embryo.

At the 8-cell stage, on the other hand, AGS/Gαi showed a critical role in nuclear migration of vegetal blastomeres to the vegetal cortex[19]. This event is distinct from AGS/Gαi's typical function in spindle orientation and positioning. Although it is yet to be identified how AGS/Gαi controls this critical event prior to asymmetric cell division, a recent report in mammalian stem cells[42] proposes that LGN (AGS ortholog) interacts with Inscuteable (Insc) to constitute stable cores of Par3-Insc-LGN-Gαi at the cortex independent of NuMA and microtubule motors. Therefore, enrichment of LGN/Gαi into the specific region of the cortex could occur independent of microtubules, prior to asymmetric cell division. This model may explain the mechanism of AGS/Gαi enrichment at the vegetal cortex prior to asymmetric cell division in the sea urchin embryo, which needs to be further tested in the future.

**Fig. 7** Sea urchin AGS induces asymmetric cell divisions during early embryogenesis and extra invaginations after blastulation in sea star embryos. **a** A summary diagram that depicts Vasa (red) and AGS (green) localization patterns during 8–16 cell stage. Sea urchin embryos undergo asymmetric cell division to form micromeres (organizers) accompanied by Vasa and AGS enrichment at the vegetal pole at 16-cell stage, whereas sea star embryos undergo symmetric cell division with no enrichment of either molecule. **b–f** Brightfield images (**b**), Live imaging (**d**), or Immunofluorescence images (**e**) of sea star embryos expressing 1.5 µg/µl stock of SpAGS-GFP mRNA that underwent random asymmetric cell divisions to form micromere-like cells during early embryogenesis (arrowheads in (**b**) and (**d**)) and extra invaginations at the larval stage (arrowheads in (**e**)). On the other hand, the controls injected with 1.5 µg/µl stock of SpAGS-dGoLoco#1 or SpAGS-dC-term that lacks GoLoco#1 or the entire C-terminal domain, respectively, or with dye underwent symmetric cell divisions (**c**) and single invagination (**f**). The number of embryos that underwent asymmetric cell divisions from 2- to 16-cell stage (**c**) or extra invagination at Day 2 (**f**) were scored and shown in %. n indicates the total number of embryos scored. In **d**, AGS-GFP was enriched on the spindle and at the cortex (arrow). White squared regions are enlarged in bottom row images. A white squared region is enlarged in the bottom row images. In **e**, Vasa (red) was enriched in the germline of the control larva (arrow), which was not found in the experimental group. n in the graphs **c** and **f**, indicates the total number of embryos scored. Each experiment was performed at least three independent times. Each image shows the representative phenotypes scored in the corresponding graph. Scale bars = 50 µm

AGS/Gαi forms a protein complex with suNuMA–Dynein–Vasa that is present on the spindle and stabilized in the vegetal cortex, resulting in Vasa enrichment in micromeres at the 16-cell stage. This AGS/Gαi-dependent machinery also appears to anchor other proteins and mRNAs important for cell fate specification to the vegetal cortex during micromere formation. Micromeres thus acquire a unique molecular status compared to other embryonic cells, directing this cell lineage to fate specification as a major signaling center which contributes to the vegetal plate specification[43,44]. In the subsequent asymmetric cell division to form the large and small micromeres, the same AGS/Gαi-dependent machinery also appears to be reutilized: Vasa and its associated factors all accumulate toward the vegetal, small micromere-side during the asymmetric cell division. It is unclear though how the micromere descendants differentially distribute developmental factors for large (skeletogenic cell) or small micromere (primordial germ cells) lineage during this asymmetric cell division. Since the second asymmetric cell division results in the small micromeres losing contact with the overlying embryonic cells, perhaps reciprocal signals change the large and small micromeres unevenly. Alternatively, the small micromeres contain the vegetal pole-most material and may influence fate distinctly. Further, when micromeres are removed, Vasa becomes overexpressed elsewhere in the entire embryo yet, is eventually restricted to the germline. Embryos could thus re-establish a germline by larval stage[24,45]. It is yet to be tested if/how AGS-mediated machinery contributes in re-establishment of the vegetal lineages, including a germline, in the micromere-depleted embryos.

A feedback loop may amplify an initial cortical asymmetry leading to the observed phenotype. In HeLa cells, PLK1 dissociates Dynein/Dynactin from the NuMA-LGN complex to control the microtubule pulling force at the cortex[30]. We found in the sea urchin embryo a similar regulatory mechanism for its asymmetric cell division. A difference in the sea urchin mechanism is that PLK1 appears to dissociate not just Dynein/Dynactin but also suNuMA from AGS/ Gαi at the cortex. This may be due to molecular differences in AGS versus its human ortholog LGN or between human NuMA and suNuMA, which needs to be tested in the future. Further, in the sea urchin embryo, cortical AGS/Gαi localization was maintained through several cell divisions (from 4–32 cell stage), which is distinct from other cells (such as HeLa cells) where it disappears each cell cycle[30]. It is, however, unclear at this point what regulates this sustained cortical AGS localization through multiple cell divisions. Since another polarity factor Dishevelled (Dsh) is maternally polarized at the vegetal cortex prior to fertilization[46], it is important to test in the future if/how Dsh and AGS may function together in this process of cortical asymmetry establishment.

The N-terminus of AGS/LGN is conserved among diverse animals yet the C-terminus is highly variable. We demonstrated

that the sea urchin AGS could directly recruit and anchor spindle poles to the cortex, which is a distinct event for micromere formation compared to other cell divisions in the sea urchin embryo or in other echinoderms that all undergo symmetric cell divisions during early embryogenesis. This indicates that cortical AGS has an essential contribution to micromere formation. Intriguingly, AGS localization in the vegetal cortex was observed in pencil urchin embryos also—but only when micromeres are formed. This experiment suggests a close link between the cortical AGS localization and acquisition of micromeres in an early onset of echinoid diversification in both an observation of nature, and in an experimental test. Further, sea urchin AGS (SpAGS) induced spindle pole anchoring as well as micromere-like cell formation not just in the sea urchin embryo but also in the sea star embryo where cortical AGS is normally absent and no asymmetric cell division occurs during early embryogenesis. Because the protein sequences of sea urchin AGS and sea star AGS are almost identical except in the C-terminus (Supplementary Table 5), these results point to evolutionary modifications of AGS in the C-terminus with an additional GoLoco motif. This simple change in SpAGS might have provided stronger pulling force by the GAND complex at the vegetal cortex, causing the sustained spindle anchoring and formation of micromeres in echinoid species.

An extra GoLoco motif may also lead to a conformational change and/or activity of the AGS protein, precipitating its new molecular influence. It is known that Pins/AGS is generally present in a closed/auto-inhibited state because of intramolecular binding between its N-terminal TPR domains and C-terminal GoLoco motifs. In *Drosophila*, in this inactivated form, only the GoLoco motif #1 that is topologically located outside of TPR-binding stays active and functions in recruiting Pins to the cortex[39]. Once Pins reaches the cortex, it is then opened and activated by binding of Gαi to the rest of GoLoco motifs for spindle regulation[47]. *Drosophila* Pins has three GoLoco motif #1–3, and its ortholog human LGN and sea urchin AGS have four GoLoco motif #1–4. Sequences of the GoLoco motifs are more similar among each GoLoco motif of various organisms than among different GoLoco motifs of the same organisms (Supplementary Table 7b, c). Therefore, one can predict which GoLoco motif is present or absent in each organism. We identified that the GoLoco motif #1 is present only in echinoid species among echinoderms (Fig. 9a; Supplementary Table 7a; see also Methods). Therefore, in the sea urchin embryo, this GoLoco motif #1 may have provided recruiting activity of AGS to the cortex in addition to providing a more sustainable pulling force at the vegetal cortex,

In this report, we also demonstrated that SpAGS is sufficient to induce asymmetric cell divisions and potentially to facilitate vegetal fate behavior and polarity. Sea star embryos are much less feasible for embryological manipulations compared to sea urchin

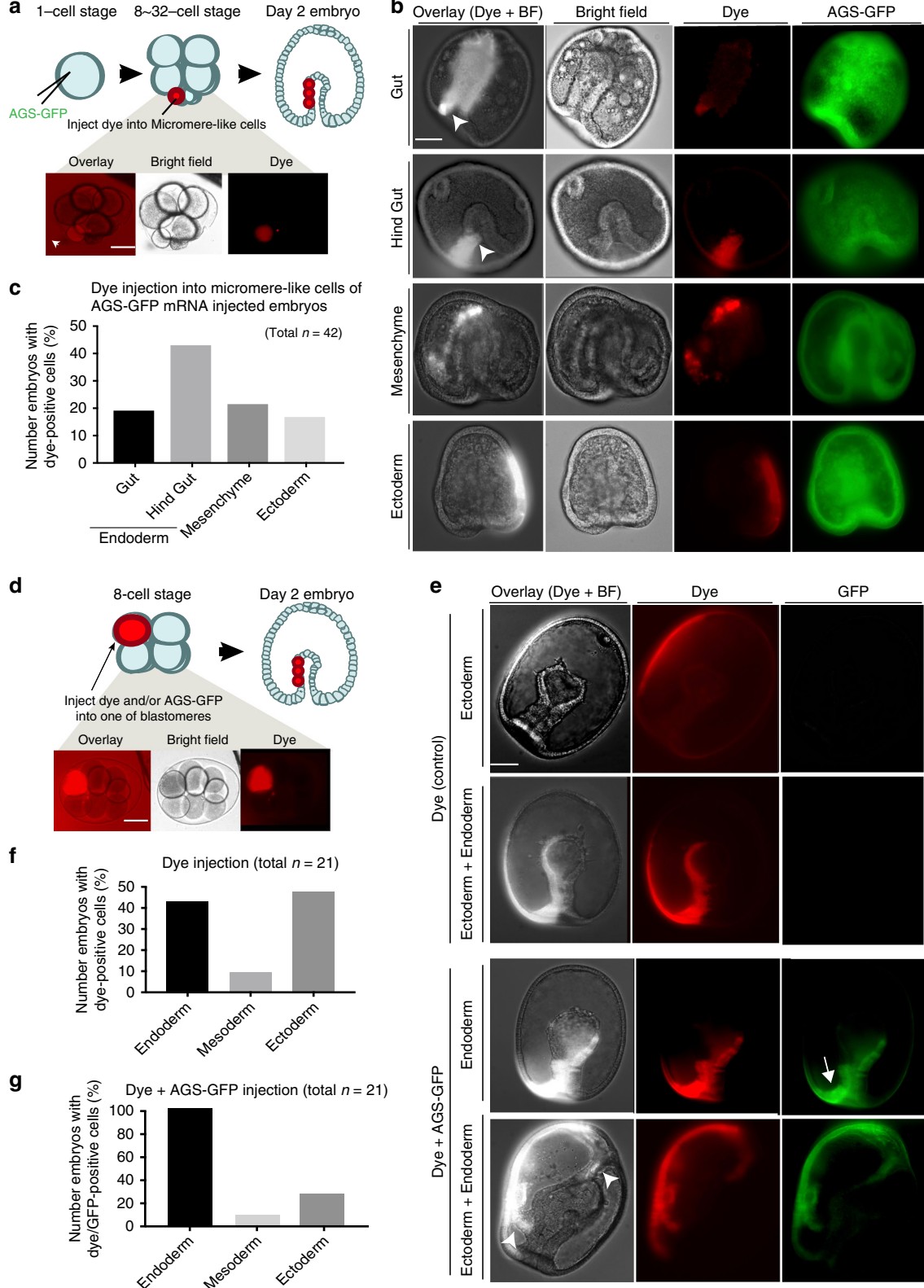

embryos, and thus a conventional blastomere transplant experiment such as micromere transplant is currently technically not feasible in the sea star embryo. We, therefore, directly introduced SpAGS mRNA into one of eight blastomeres to mimic formation of micromeres in the sea star embryo. Under this condition, many embryos showed somewhat abnormal development likely due to random induction of polarity by SpAGS. To fully recapitulate the

functional micromeres or the developmental program of the sea urchin in the sea star embryos, more targeted approaches such as optogenetics that allows a spatio-temporal control of protein regulation are needed in the future.

Overall, we propose that the ancestral sea urchin modified an evolutionarily conserved AGS-dependent regulatory mechanism to induce asymmetric cell divisions in its early embryo, which has

**Fig. 8** SpAGS-overexpression induces endo-mesodermal lineages in sea star embryos. **a–c** 1.5 µg/µl stock of SpAGS-GFP mRNA was injected into unfertilized eggs and the micromere-like cells were labeled with red fluorescent dextran (dye) during 8–32 cell stage (**a**). At Day 2, embryos were scored for each category based on where the dye signal was found (**b**, **c**). Arrows indicate the signal enrichment at the bottom of the gut. At the occasion when the signal was scattered in various tissues, the embryo was counted for each category. n indicates the total number of embryos scored. **d–g** 0.75 µg/µl stock of SpAGS-GFP mRNA or dye was injected into one of the blastomeres at the 8-cell stage (**d**). At Day 2 (**e**), embryos that were injected with dye (**f**) or with dye and SpAGS-GFP mRNA (**g**) were scored for each category based on where the dye signal was found. When the signal was scattered in various tissues, the embryo was scored in each lineage category. In **e**, arrow indicates AGS-GFP enrichment at the vegetal-most region of the gut and arrowheads indicate the signal enrichment of extra invaginations. *n* in the graphs **f–g** indicates the total number of embryos scored. Each experiment was performed at least three independent times. Each image shows the representative phenotypes scored in the corresponding graph. Scale bars = 50 µm

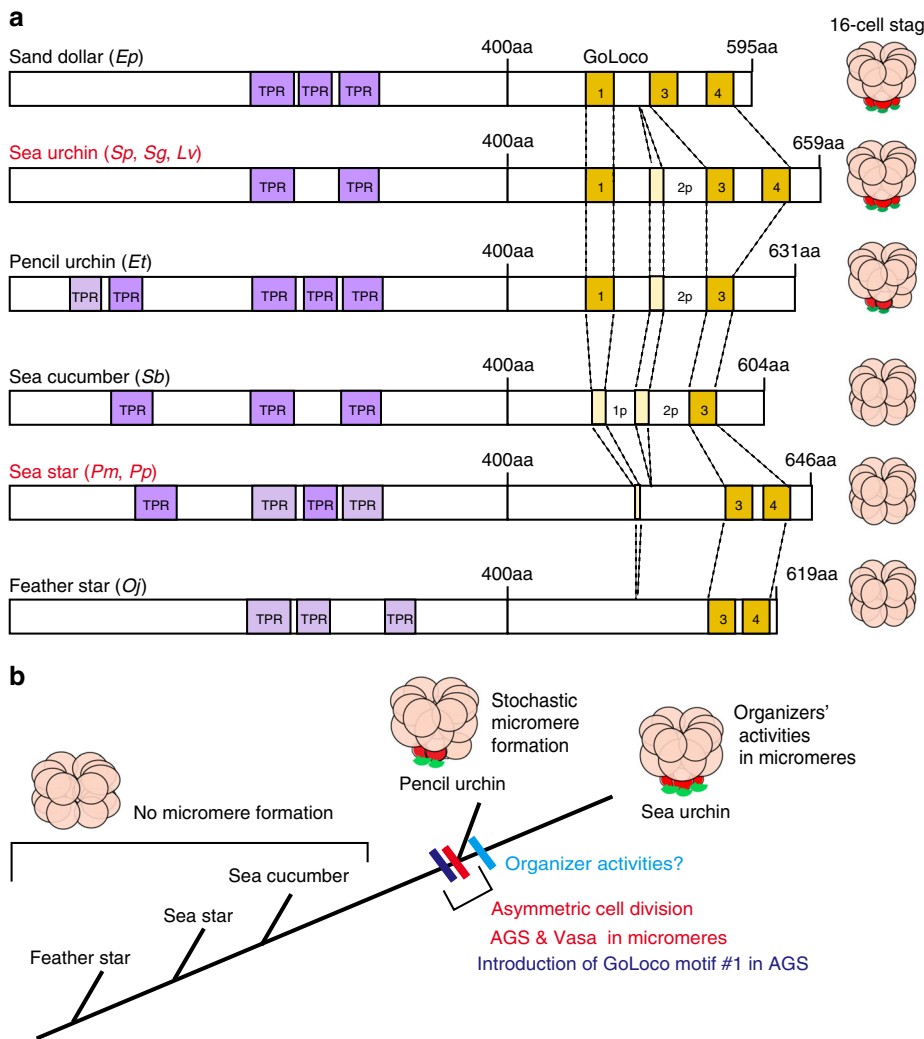

**Fig. 9** The evolutionary modification of AGS proteins among echinoderms. **a** Predicted motifs of each AGS protein are depicted based on NCBI blast search results (See Supplementary Tables. 6 and 7 for details). Conserved TPR domains are highlighted in purple and GoLoco motifs in yellow (See Supplementary Table 7B for the specific sequence of each GoLoco motif). Less conserved or partial domains are colored in light purple or light yellow, respectively. ( ) indicates species used to construct each cartoon. Vasa, red; AGS, green. **b** A model for introduction of micromeres during echinoid diversification. The timing of micromere formation coincides with introduction of GoLoco motif#1 in AGS protein during echinoid diversification

in part contributed to its distinct cell interactions and developmental style among echinoderms (Fig. 9b). This is a mechanism likely used by many cells in various embryos and organisms that acquired new cell types and signaling activities at some point during the course of evolution to establish more rapid, maternally inherited developmental guidance.

## Methods
**Animals**. *Strongylocentrotus purpuratus* (sea urchins) and *Patria miniata* (sea star) were collected from the ocean by Pat Leahy, Kerchoff Marine Laboratories,

California Institute of Technology or by Josh Ross, South Coast Bio- Marine LLC. Long Beach, California, USA, and maintained in the cooling aquarium at 16 °C. *Eucidaris tribuloides* (pencil urchins) were collected from the ocean by Ken Nedimyer Florida Keys and maintained in the aquarium at room temperature.

**Handling of echinoderm embryos**. Gametes were acquired by 0.5 M KCl injection (sea urchins and pencil urchins) or dissection (sea star). Eggs were collected in seawater (SW) and sperm were collected dry. To obtain embryos, fertilized eggs of *S. purpuratus* and *P. miniata* were cultured in SW at temperatures at 16 °C. When early stage embryos were required for fixation, fertilization was performed in the presence of 1 mM 3-aminotriazol (Sigma, St. Louis, MO, USA) to remove fertilization envelopes. Pencil urchin embryos were fertilized and cultured at room

temperature. Prior to fixation, the fertilization envelopes were damaged slightly to allow permeabilization.

**In situ RNA hybridization and Immunofluorescence.** For in situ hybridization, RNA probes for *Sp-vasa*, *Sp-seawi*, *Sp-delta*, and *Sp-cyclinB* were previously constructed[45–48]. Briefly, a DIG RNA Labeling Kit (T7) (Roche, Indianapolis) was used to construct antisense DIG-labeled probes to each target mRNA from a cDNA template, and the DIG-labeled RNAs were then hybridized to embryos for one week and detected by TSA Plus Fluorescence Systems (PerkinElmer, cat# NEL760001KT)[49,50]. For Immunolabelling, embryos were fixed with 90% methanol or 3.7% PFA with 0.1% triton-X-100, washed with PBS and treated with antibodies[24]. The primary antibody was used at final concentration as follows: Anti-SpAGS at 1:500[25], Anti-SpVasa at 1:300[24], Anti-Gαi at 1:30 (sc-56536, Santa Cruz Biotech), Anti-suNuMA at 1:500, Anti-human PLK1 at 1:100 (ab137352 or ab14210, Abcam), Anti-Endo1 at 1:10[51], or FITC conjugated Anti-beta-tubulin at 1:50 (F2043, Sigma). The secondary antibodies Cy3, Alexa488 and Alexa594 (Invitrogen) were used at 1:250, 1:500 and 1:500, respectively. Hoechst was used at a final concentration of 0.1 μg/ml. Stained embryos were imaged on a wide-field fluorescence microscopy (Zeiss Axioplan), or on a Zeiss LSM 510 Meta Confocal Laser Scanning Microscope or a Zeiss LSM 800 Confocal Laser Scanning Microscope and analyzed using Zen software. The details of the reagents are also listed in Supplementary Table 8.

**Chemical treatment.** For chemical treatment, embryos were treated with final 2 μg/mL PTX for Gαi inhibition[52] (P2980, Sigma), 25 μg/mL SDS to disrupt asymmetry, 100 μM Ciliobrevin A to inhibit cytoplasmic AAA + ATPase dynein-dependent microtubule gliding and ATPase activity[53] (TOCRIS bioscience), or 10 μM BI2536 to block PLK1 activity (Selleckchem, USA) during the 8–16 cell stage for 10 min before fixation for immunofluorescence. For phalloidin staining, embryos were fixed with 3.7% paraformaldehyde for 15 min, followed by PBS wash three times with 0.1% triton, incubated with 10 unit/mL of Rhodamine-Phalloidin (Thermo Fisher Scientific) for 20 min, washed with PBS three times all at room temperature prior to imaging. The details of the reagents are also listed in Supplementary Table 8.

**suNuMA antibody designing and production.** C-terminus of eIF3s5L (SPU_023686) was used to produce a peptide antibody. This molecule was first annotated as eIF3s5L with some sequence similarity to eIF3s5L in the first 200aa N-terminus of the total 1886aa protein. The rest of its sequence, however, showed no similarity to eIF3s5L and its C-terminus was most similar to vertebrate NuMA-like proteins. The sequence was the most similar in the region of the predicted microtubule-binding site. Antibodies were designed within this putative microtubule-binding domain because this region is most conserved with human NuMA and other deuterostomes including a hemichordate and vertebrates. Sequences used for antibody production are #1 DRLAELQRRNTL, #2 CRPHLQTSYPVETQ and #3 TRAPKEITETDL. Due to a potential breakdown of a protein, a ladder or smear of small protein bands were sometimes found, a major band of NuMA protein was detected around 210 kb with each of these antibodies as predicted. #1 antibody was used for all analyses in this paper.

**mRNA Injection and microscopy.** PLK1-GFP mRNA was used to see specific PLK1 protein localization, membrane-mCherry-PLK1 and membrane-mCherry-PLK1-Kinase Dead (PLK1-dead) mRNAs to test PLK1 activity[30]. Human AGS-GFP (#AGS03L0000) was obtained from Missouri S&T cDNA resource center, and human LGN-GFP mRNAs was also previously used[30]. Vasa-GFP mRNA to visualize Vasa protein localization was used previously in Yajima and Wessel (2011b, 2012, 2015)[10,16,45]. For SpAgs-mRNA and SpGαi-mRNA construction, SpAgs or SpGαi- ORF was amplified by PCR using primers listed in Supplementary Table 8, and subcloned into the pSp6 β-globin UTR plasmid between the Xenopus β-globin 5′ and 3′ UTRs for in vitro transcription[54]. The GFP or Kaede that was amplified by PCR from the Kaede-centrin1 template[55] was fused either to the C-terminus of SpAgs or the N-terminus of SpGαi using In-Fusion HD Cloning kit by following manufacturer's protocol (#639648, Clontech, USA). To remove the GoLoco motif #1 (473aa DNFFEALSRFQSNRMDEQR CSF 495aa) from SpAGS-GFP, the internal *BbvC1* (458a) and *Bsm*1 (532aa) sites were used to remove the sequence including GoLoco motif #1, and the corresponding sequence that lacks only the GoLoco motif #1 was fused back using In-Fusion HD Cloning kit described above. To remove the entire C-terminus of AGS, the internal Spe1 sites were used to remove the 394aa-658aa of AGS and the vector was self-ligated. mCherry-EMTB was obtained from Addgene (#26742)[56].

Unless individually indicated, embryos were in general injected at the one cell stage with 0.25–0.5 μg/μl of AGS-GFP and GFP-Gαi, 0.4–0.5 μg/μl of mCherry-EMTB mRNA, 1 μg/ul stock of PLK1-GFP, or lower dose (0.25 μg/ul stock) of membrane-targeted PLK1 or PLK1-dead for mild perturbation. For sea star embryo injection, ~1.5 μg/ul of AGS-GFP, AGS-dGoLoco1-GFP, or AGS-dC-term-GFP was used due to the larger size of the egg (200 μm diameter) compared to the sea urchin one (80 μm diameter). Embryos were imaged either on a wide-field fluorescence microscopy (Zeiss Axioplan) or a Zeiss LSM 800 Confocal Laser

Scanning Microscope. For live imaging, they were imaged for approximately 15 min with 30-s intervals.

**Morpholino preparations and testing.** A morpholino antisense oligonucleotide (MO) that specifically blocks translation of the targeted molecule was designed as complementary to the 5′ UTR each of Sp-Ags, Sp-Gαi, Sp-NuMA (eIF3s5L), Sp-Plk1 (GeneTools, Philomath, OR) and each sequence is listed below. Each of these MO was tested for a possible non-specific toxicity by injecting each into sea star embryos prior to use in this study. Only ones that showed no-toxicity in sea star embryos were used in this study. As controls, Sp-Nanos2[57] or Sp-Dsh that should not show any phenotype at 16-cell stage was always used as controls alongside with each MO experiment described above. The level of MO knockdown was tested by the signal reduction in immunofluorescence and/or immunoblotting. For injection, MOs were prepared at the stock concentration of 2 mM each unless specifically indicated in the text and rescue mRNAs to test for specificity were prepared at the stock concentration of 0.25–1 μg/μl, and 6pl each of the stock solution was injected into fertilized eggs. Fluorescent images were taken by confocal laser microscopy (Zeiss LMS510 or LMS800) or wide-field fluorescence microscopy (Zeiss Axioplan). List of Genomic loci, primers, and morpholino target sites are also listed in Supplementary Table 8.

**Blastomere manipulation.** For micromere transplants, donor embryos were pre-stained with 5 μg/ml RITC (Sigma) for one hour in advance of the transplantation at 16-cell stage[58,59]. The RITC-labeled micromeres (donor) were transplanted onto the unstained embryos that were previously injected with AGS-MO (host) at the 1-cell stage together with a minimum amount of green fluorescent dye to avoid any potential cytotoxicity. Any uninjected embryos were thus removed from the group immediately after injection and only injected embryos were used as host embryos for the transplantation at the 16-cell stage. As AGS-kd embryos lacked apparent micromeres at the 16-cell stage and it was difficult to identify the orientation of Animal-Vegetal axis or the exact developmental timing, donor micromeres were transplanted at the random location of the host embryos around 4.5 h post-fertilization (PF). For blastomere injection, embryos of the desired stage were loaded onto the injection chamber and vertically injected either with 1 mg/mL stock of red fluorescent dextran (Fluoro-Ruby, Invitrogen) or with a mixture of red dextran and 0.75 μg/μL stock of SpAGS-GFP-mRNA. The amount of injection was monitored in real-time under the fluorescent microscope during injection. For Kaede protein photoconversion[60,61], embryos injected with 1.5 μg/ul stock each of SpAGS-GFP and Kaede mRNAs were placed on the stage of the Olympus FV3000 confocal microscope during 8–32 cell stages and a micromere-like cell was irradiated with 12% UV laser for 0.8 s. The level of the photoconversion was monitored in real time by continuous imaging. The manipulated embryos were cultured and imaged at Day 2–3 PF. Images were taken by wide-field fluorescence microscopy (Zeiss Axioplan). The bright-field images were overlaid with the fluorescent images taken at the same focal plane.

**Immunoprecipitation and immunoblotting.** Gαi was used here for Immunoprecipitation (IP) because it showed the most specific localization at the cortex in the limited time of the asymmetric cell divisions (8–32 cell stage), whereas AGS was present in the vegetal cortex as well on the spindle of every blastomere throughout embryogenesis. 8–16 cell stage embryos were treated with 100 μM Taxol (Paclitaxel-T7402, Sigma) for 10 min to stabilize microtubules (Fig. 2a and f and Supplementary Fig. 4a), or were treated with 16 ng/mL Nocodazole (Sigma) for 10 min to inhibit the microtubule polymerization (Fig. 2b). For Fig. 2a and f experiments, one set of lysates was prepared in the presence of final 10 μM BI2536 (Selleckchem, USA), a specific inhibitor of PLK1 activity[32] and Gαi-IP was performed. The other group of lysates was IP-ed with Gαi, and then treated with recombinant PLK1 protein that has a kinase activity in vitro (PV3501, ThermoFisher Scientific).

Approximately, 6 μg each of Gαi antibody (sc56536, Santa Cruz Biotech), SpVasa antibody [24], or suNuMA antibody #1 (peptide antibody constructed by Genescript, HK) was used per sample for immunoprecipitation (IP). The IP was performed as described in the instruction manual of Dynabeads Protein A (Invitrogen). Instead of SpGαi antiserum previously constructed [26], a monoclonal mouse Gαi antibody was used in this report to reduce the background as Gαi sequence is highly conserved between sea urchin and human (Supplementary Table. 3). Half of the IP-ed samples were treated with 1 μg of PLK1 protein for 1 h at 30℃. The resultant samples were then prepared in 100 μL of loading buffer for polyacrylamide gel electrophoresis. Each sample (10 μl, for all, or 15 μl for Fig. 2b experiment) was run on a 4–20 % gradient Tris–glycine polyacrylamide gel (Invitrogen, Carlsbad, CA) and transferred to nitrocellulose membranes for immunoblotting with SpVasa antibody at 1:1000, suNuMA antibody at 1:2000, Dynein antibody (MAB1618, Millipore) at 1:500, p150 antibody (#610473, BD Bioscience, USA)[30] at 1:500, Gαi antibody at 1:200, or Phosphoserine antibody (P3430, Sigma) at 1:500, SpAgs antiserum [25] at 1:2000, and peroxidase-conjugated rabbit or mouse anti-mouse secondary antibodies (1: 5000) (ThermoFisher Scientific). For Gαi antibody detection, peroxidase-conjugated anti-Protein A antibody (ab7245, Abcam) was used as a secondary antibody because Gαi overlaps with IgG heavy chain at around 64 kDa. YP30 (York protein 30) antibody was used at 1:40,000 to normalize the signal[31]. The reacted proteins were detected by

incubation in a chemiluminescence solution (1.25 mM luminol, 68 μM coumeric acid, 0.0093% hydrogen peroxide and 0.1 M Tris pH 8.6) for 1–10 min, exposed to film and developed. Each experiment was performed at least three independent times. All un-cropped blots are enclosed in the Source data file. The details of the reagents are also listed in Supplementary Table 8.

**Data analysis**. Measurements presented in Fig. 1c-k were performed by *Image J* as described below, and the statistical significance obtained by one-way Nova between control group and experimental groups was indicated as *p*-value. A positioning of the nucleus in the vegetal blastomere at the late 8-cell stage (Fig. 1c, d): The distance was measured from the vegetal pole to the edge of the nucleus (vd) or to the cellular outline (td). The relative value was obtained by value (vd) divided by value (td). Spindle orientation and centrosome positioning were measured as follows (Fig. 1e, f): The angle of spindle (the micromere-side of centrosome to the macromere-side of centrosome) was measured by setting the vegetal pole plane at 0 degree and measuring the intersection point of the spindle (r). The centrosome positioning was measured by the distance from the vegetal pole to the micromere-side of centrosome (vd) or to the end of cell wall (td). The relative value was obtained by value (vd) divided by value (td). PTX treatment was performed only for anaphase embryos. Unequal cell division was measured as follows (Fig. 1g, h): Relative size of the micromere to the adjacent macromere was analyzed by measuring diameter of each cell at the 16-cell stage (Micro/ Macro). Vasa asymmetric distribution was measured as follow (Fig. 1i, j): The relative signal intensity was measured on the micromere-side of spindle and compared to that on macromere-side of spindle during anaphase of the 8–16 cell stage (Micro-side/ Macro-side). Vasa enrichment in the vegetal pole (Fig. 1k) was measured by manually counting the number of embryos with the Vasa signal under the microscope at 16-cell stage. Each experiment was repeated at least three independent times.

**Blast and motif analysis**. All echinoderm AGS sequences were obtained from *Echinobase.org* through a blast search against SpAGS (SPU_009218). Protein sequence alignment and molecular phylogenetic tree was constructed by *crustal Omega*. Protein structural motif analysis was performed through the NCBI blast search with the value threshold of 0.005, a slightly more stringent threshold than the default to remove further non-specific hits.

**GoLoco motif analysis**. GoLoco motif is a protein structure with three alpha helices and each of the motifs consists of different amino acid sequences[62]. In Fig. 9a, each GoLoco motif was numbered based on the sequence similarity to that of other echinoderms. For example, GoLoco motif #1 of sea urchins is more similar to the GoLoco motif # 1 of other echinoderms than GoLoco motifs #2–4 of the same sea urchin species (Supplementary Table 7b).

**Reporting summary**. Further information on research design is available in the Nature Research Reporting Summary linked to this article.

## Data availability
The authors declare that all data supporting the findings of this study are available within the article and its supplementary information files or from the corresponding author upon reasonable request. The source data underlying Figs. 1, 2a, b, f, 4c, 5c, 7c, f, 8f, g, and Supplementary Figs. 3a, 4 and 7a are provided as a Source Data file.

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

## Acknowledgements
The authors thank Dr. Tomomi Kiyomitsu at Nagoya University for sharing reagents and for critical discussion, and members of the PRIMO at Brown University for generous support. This work was supported by the American Heart Association Scientist Development Grant (14SDG18350021) and NIH (1R01GM126043-01) to MY, and by NIH (2R01HD028152) and NSF (IOS-1120972) grants to GMW.

## Author contributions
J.P. and A.F. were responsible for experimental design and undertaking, data analysis, manuscript construction; G.M.W. was responsible for experimental design, manuscript editing; M.Y. was responsible for concepts, experimental design and undertaking, data analysis, manuscript construction and editing.

## Additional information

**Competing interests:** The authors declare no competing interests.

