## [Peer Review File · Nature Communications]

Reviewers' Comments:

Reviewer #1:

Remarks to the Author:

In this MS, the authors explore the molecular mechanisms which control asymmetric division and fate specifications in Sea urchin micromeres. They find that AGS (an homologue of PINS) function with Gai and NuMa to recruit dynein activity yielding to asymmetric division. This system also mediates the translocation of VASA complex to the small micromeres by locally regulating PLK1. The paper then goes on to test what modification in AGS may account for the loss of asymmetric divisions in other echinoderms like pencil urchins and sea stars; thereby relating domain evolution of a conserved factor with a key developmental feature. Although not extremely new at the molecular level, the findings reported bring significant new advances in addressing the mechanisms for asymmetric divisions in vegetal sea urchin blastomeres, which has long served as a striking example for the role of asymmetric division in fate specification and development. The evolutionary findings are also quite exciting. On a more negative note, the paper suffers from its density and presentation, and is difficult to read; with figures panels difficult to understand. Although I have no specific request for additional experiments, I would like the authors to make a significant effort in the presentation.

Some specific points:

- The flow of the paper would benefit from some reorganization. For instance, I did not understand why Figure 2 was presented here. To me the logic, is to first address mechanisms of asymmetric divisions, and asymmetric fate segregation (Fig 1, 3,4), and then delve into the implications in development (Fig 2) and evolution (5 and 6).

- Some panels are not clearly presented or poorly conveying the messages claimed in the text. Here are some examples: 1C, the single channels are zoomed as compared to the merge, with no scale bars. In Figure 1F, the Gai KD division appears symmetric, and in both the Gai KD and PTX treatment the asymmetric localization of VASA is barely convincing. Figure 4B, the embryos appear to have formed smaller cells through an asymmetric division, thus it is not clear how the authors interpret the lack of micromere formation in 85% of embryos injected with membrane-PLK1. Fig S5 D how did spindles formed in the absence of dynein activity?

- In the discussion the authors state that AGS/ Gai KD has no effect on A-V oriented divisions at the 4-cell stage, but how can they ensure that the Morpholino is effective there? In general a cleaner quantification of the time-dependent depletion associated with there MO would help. Also there are interesting theoretical and experimental analysis of division orientation / positioning in Pierre at al. Developmental Cell 2016, which suggest that the elongated shape of 4 cell blastomeres in urchins may be sufficient to align the spindle along the A-V axis independent of cortical effectors. This work should be discussed. Finally, another peculiarity of this asymmetric division is that it first involves an asymmetric displacement of nuclei, then followed by spindle assembly; this is an interesting difference as compared to systems for spindle positioning implicating similar cortical effectors, and shall also be discussed.

Reviewer #2:

Remarks to the Author:

This manuscript is a new in-depth account of how micromeres are formed in echinoids breaking new ground on the mechanisms that generate asymmetrical cleavage, an important phenomenon in developmental biology. The details are meticulously presented and the results are solid.

How AGS, with an additional motif in echinoids is located at the vegetal cortex and the positions micromere spindles near the vegetal pole, is set out with excellent images and image analysis. The data are extensive and fully support the hypotheses being tested. The role of AGS is tested with

AGS inhibitors. The claim that small micromeres are an organizing centre adds to their previously suggested restricted fate, that of being primary germ line cells. The results with asteroid embryos are very interesting, but one wonders if the time lines are a bit different and that they should have also looked a later stage asteroid embryos.

The approach to the research is comprehensive with application of many leading-edge methods. There is much to read in this contribution, but it will be essential reading for those working in the micromere field. Others interested in the evolution of asymmetrical cell division will also find this an important paper.

By virtue of being such a data rich manuscript it also made the reading challenging as so much information is being conveyed within the confines of the journal format. I wonder if a summary table will help - with hypotheses on the left and findings in support/or not on the right.

Corrections

Please use the prime symbol (´) for related panel images and not the apostrophe.

p. 1 change "evolutionarily introduced" to "evolutionary novelty"

They cannot say throughout the echinoderm phylum as they did not test the sister class to the echinoids – the holothuroids.

Excellent work

Reviewers' comments:

Reviewer #1 (Remarks to the Author):

In this MS, the authors explore the molecular mechanisms which control asymmetric division and fate specifications in Sea urchin micromeres. They find that AGS (an homologue of PINS) function with Gai and NuMa to recruit dynein activity yielding to asymmetric division. This system also mediates the translocation of VASA complex to the small micromeres by locally regulating PLK1. The paper then goes on to test what modification in AGS may account for the loss of asymmetric divisions in other echinoderms like pencil urchins and sea stars; thereby relating domain evolution of a conserved factor with a key developmental feature. Although not extremely new at the molecular level, the findings reported bring significant new advances in addressing the mechanisms for asymmetric divisions in vegetal sea urchin blastomeres, which has long served as a striking example for the role of asymmetric division in fate specification and development. The evolutionary findings are also quite exciting.

On a more negative note, the paper suffers from its density and presentation, and is difficult to read; with figures panels difficult to understand. Although I have no specific request for additional experiments, I would like the authors to make a significant effort in the presentation.

- We carefully re-organized the figures and the text to improve the clarity.

Some specific points:

- The flow of the paper would benefit from some reorganization. For instance, I did not understand why Figure 2 was presented here. To me the logic, is to first address mechanisms of asymmetric divisions, and asymmetric fate segregation (Fig 1, 3,4), and then delve into the implications in development (Fig 2) and evolution (5 and 6).

- We revised accordingly.

- Some panels are not clearly presented or poorly conveying the messages claimed in the text. Here are some examples:

Figure 1C, the single channels are zoomed as compared to the merge, with no scale bars.

Figure 1F, the Gai KD division appears symmetric, and in both the Gai KD and PTX treatment the asymmetric localization of VASA is barely convincing.

Figure 4B, the embryos appear to have formed smaller cells through an asymmetric division, thus it is not clear how the authors interpret the lack of micromere formation in 85% of embryos injected with membrane-PLK1.

Fig S5 D how did spindles formed in the absence of dynein activity?

- We added more explanations in the main text and figure legends for each of the points raised. As for 1F, GaiKD has only mild affect on unequal cytokinesis yet Vasa distribution at the vegetal cortex was still decreased as shown in the figure (dashed circle), so we were trying to make a point that the molecular enrichment (represented by Vasa here) may be still compromised even though unequal cytokinesis was successful in these embryos. As for 4B, 85% was calculated based on Vasa enrichment but not by the size of the cells. For S5D, Embryos were treated with the Dynein inhibitor only for 10 minutes prior to fixation, so the cells were already in M-phase.

- In the discussion the authors state that AGS/ Gai KD has no effect on A-V oriented divisions at the 4-cell stage, but how can they ensure that the Morpholino is effective there? In general a cleaner quantification of the time-dependent depletion associated with there MO would help.

- This is a good point and we amended our statement in the text.

From our past experiences, we know MO can be effective from the 2nd cell division for other molecules (e.g. Vasa, CyclinB), but the reviewer is correct that the effectiveness of MO will depend on the levels of both the active maternal mRNAs and proteins of each molecule. As for quantification, MO does not affect the maternal proteins, thus it is very difficult to identify the level of depletion (the maternal load is huge and masks the zygotic signal). Further, not all maternal proteins are active in embryonic cells. Calculating only the active subset of maternal proteins is not feasible in the current field of embryology.

Also there are interesting theoretical and experimental analysis of division orientation / positioning in Pierre et al. Developmental Cell 2016, which suggest that the elongated shape of 4 cell blastomeres in urchins may be sufficient to align the spindle along the A-V axis independent of cortical effectors. This work should be discussed.

- The paper reviewer suggested is listed below but seems not relevant.

Nucleus to Mitochondria: Lost in Transcription, Found in Translation.

St-Pierre J, Topisirovic I.
Dev Cell. 2016 Jun 20;37(6):490-2. doi: 10.1016/j.devcel.2016.06.003.

-We wonder the reviewer meant another paper published in Cell? We cited and discussed about this paper in the main text.

Influence of cell geometry on division-plane positioning.

Minc N, Burgess D, Chang F. Cell. 2011 Feb 4;144(3):414-26. doi: 10.1016/j.cell.2011.01.016.

Finally, another peculiarity of this asymmetric division is that it first involves an asymmetric displacement of nuclei, then followed by spindle assembly; this is an interesting difference as compared to systems for spindle positioning implicating similar cortical effectors, and shall also be discussed.

- The upstream regulators of AGS are not yet identified in this organism, so we can only speculate here.

Recently published article in mammalian cells proposes a new model (Culurgioni et al, Nat Comm, 2018): LGN (AGS ortholog in humans) interacts with Insc and Par3 proteins independent of NuMA prior to spindle anchoring, which may allow LGN to be extra-accumulated into the specific region of the cortex necessary for asymmetric cell division. Considering AGS/Gai enrichment at the cortex starts well before spindle anchoring occurs, this model may potentially explain. We thus cited this paper in the main text.

Reviewer #2 (Remarks to the Author):

This manuscript is a new in-depth account of how micromeres are formed in echinoids breaking new ground on the mechanisms that generate asymmetrical cleavage, an important phenomenon in developmental biology. The details are meticulously presented and the results are solid.

How AGS, with an additional motif in echinoids is located at the vegetal cortex and the positions micromere spindles near the vegetal pole, is set out with excellent images and image analysis. The data are extensive and fully support the hypotheses being tested. The role of AGS is tested with AGS inhibitors. The claim that small micromeres are an organizing centre adds to their previously suggested restricted fate, that of being primary germ line cells.

The results with asteroid embryos are very interesting, but one wonders if the time lines are a bit different and that they should have also looked a later stage asteroid embryos.

- This is an interesting future question to be addressed. In this work, the sea star embryos injected with SpAGS were in general unhappy likely due to uncontrolled polarity induction elsewhere and did not survive well longer. Thus, we will need a tempo-spatial regulation of SpAGS to fully mimic organizer formation in the sea star embryo. This point is included in the discussion section.

The approach to the research is comprehensive with application of many leading-edge methods. There is much to read in this contribution, but it will be essential reading for those working in the micromere field. Others interested in the evolution of asymmetrical cell division will also find this an important paper.

By virtue of being such a data rich manuscript it also made the reading challenging as so much information is being conveyed within the confines of the journal format. I wonder if a summary table will help - with hypotheses on the left and findings in support/or not on the right.

- We enlarged the summary figures in Fig. 3, but are uncertain if that is what the reviewer meant?

Corrections

Please use the prime symbol (´) for related panel images and not the apostrophe.

p. 1 change "evolutionarily introduced" to "evolutionary novelty"

They cannot say throughout the echinoderm phylum as they did not test the sister class to the echinoids – the holothuroids.

- Thank you for pointing out, we have amended those errors.

Excellent work

Maria Byrne

Reviewers' Comments:

Reviewer #1:

Remarks to the Author:

The authors have successfully addressed my request. The paper reads better with the novel organization. The paper I was referring to is:

"Generic Theoretical Models to Predict Division Patterns of Cleaving Embryos." *Dev Cell*. 2016 Dec 19; 39(6):667-682. doi: 10.1016/j.devcel.2016.11.018. PMID: 27997824

It is relevant to this study, as it discusses the integration of vegetal polar domains and cell shape in urchin development.

Response to Reviewers

Reviewer #1 (Remarks to the Author):

The authors have successfully addressed my request. The paper reads better with the novel organization. The paper I was referring to is:

"Generic Theoretical Models to Predict Division Patterns of Cleaving Embryos." Dev Cell. 2016 Dec 19;39(6):667-682. doi: 10.1016/j.devcel.2016.11.018. PMID: 27997824

It is relevant to this study, as it discusses the integration of vegetal polar domains and cell shape in urchin development.

- Thank you - we included this reference into the final version of our manuscript.